# Polymorphism and Ligand Binding Modulate Fast Dynamics of Human Telomeric G-Quadruplexes

**DOI:** 10.3390/ijms24054280

**Published:** 2023-02-21

**Authors:** Luca Bertini, Valeria Libera, Francesca Ripanti, Francesca Natali, Marco Paolantoni, Andrea Orecchini, Alessandro Nucara, Caterina Petrillo, Lucia Comez, Alessandro Paciaroni

**Affiliations:** 1Department of Physics and Geology, University of Perugia, Via Alessandro Pascoli, 06123 Perugia, Italy; 2CNR-IOM c/o Department of Physics and Geology, University of Perugia, Via Alessandro Pascoli, 06123 Perugia, Italy; 3CNR-IOM and INSIDE@ILL c/o OGG, 71 avenue des Martyrs, CEDEX 9, 38042 Grenoble, France; 4Department of Chemistry, Biology and Biotechnology, University of Perugia, Via Elce di Sotto 6, 06123 Perugia, Italy; 5Department of Physics, Sapienza University of Rome, P.le Aldo Moro 5, 00185 Roma, Italy

**Keywords:** G-quadruplex, BRACO19, anticancer drugs, neutron scattering, infrared spectroscopy

## Abstract

Telomeric G-quadruplexes (G4s) are promising targets in the design and development of anticancer drugs. Their actual topology depends on several factors, resulting in structural polymorphism. In this study, we investigate how the fast dynamics of the telomeric sequence AG${}_{3}$(TTAG${}_{3}$)${}_{3}$ (Tel22) depends on the conformation. By using Fourier transform Infrared spectroscopy, we show that, in the hydrated powder state, Tel22 adopts parallel and mixed antiparallel/parallel topologies in the presence of K${}^{+}$ and Na${}^{+}$ ions, respectively. These conformational differences are reflected in the reduced mobility of Tel22 in Na${}^{+}$ environment in the sub-nanosecond timescale, as probed by elastic incoherent neutron scattering. These findings are consistent with the G4 antiparallel conformation being more stable than the parallel one, possibly due to the presence of ordered hydration water networks. In addition, we study the effect of Tel22 complexation with BRACO19 ligand. Despite the quite similar conformation in the complexed and uncomplexed state, the fast dynamics of Tel22-BRACO19 is enhanced compared to that of Tel22 alone, independently of the ions. We ascribe this effect to the preferential binding of water molecules to Tel22 against the ligand. The present results suggest that the effect of polymorphism and complexation on the G4 fast dynamics is mediated by hydration water.

## 1. Introduction

Guanine-rich nucleic acid sequences are able to form non canonical DNA and RNA structures called G-quadruplexes (G4s). These structures are characterized by the stacking of planar arrangements consisting of four guanine bases linked via Hoogsteen hydrogen bonds, called *G-tetrads* [1,2,3,4]. The remaining nucleotides take part in the formation of *loops* connecting the tetrads. A fundamental role in the stabilization of G4 structures is played by the presence of monovalent cations, such as K${}^{+}$ and Na${}^{+}$, coordinated to the guanine O6 atoms in the tetrads [5,6,7].

Putative G4-forming sequences found in telomeric regions and oncogene promoters are connected to a number of vital biological processes, such as genome stability, cancer, and aging [1,8,9,10]. Telomeres comprise the d(TTAGGG)${}_{n}$ tandem repeats and vary from 5 to 25 kbases in length, with a single-stranded overhang (which allows G4 formation) of a few hundred bases [11,12,13]. G4 formation and stabilization in these regions have been proposed to inhibit the telomerase reverse-transcriptase enzyme, which is up-regulated in 85% of cancer cells [14,15]. For this reason, human telomeric G4s have attracted much interest and are, at present, regarded as promising targets for anticancer drugs [16,17]. Indeed, small molecules interacting with G4s have been found to display anticancer activity [1,18,19]. Therefore, a deeper understanding of G4-ligand interactions is key to design more effective and selective molecules.

G4s display structural *polymorphism*, which is embodied in the large variety of topologies that can be adopted depending on many factors, such as guanine number of strands, loop geometry, and strand direction. The most physiologically relevant G4 is the intramolecular, i.e., the monomeric one. Furthermore, G4s can be classified as *parallel*, *antiparallel*, and *hybrid* depending on strand orientation and glycosidic bond angle [20,21,22]. Regarding the loop arrangement, antiparallel G4s can be divided into *chair* or *basket-type* structures. The environment is a key factor in determining G4 conformational properties. It is well known that, depending on the particular G4-forming sequence, interactions with different ions, as well as different levels of hydration, may lead to different conformations [5,6,23].

Tel22, which is the prototypical human telomeric G4-forming sequence AG${}_{3}$(TTAG${}_{3}$)${}_{3}$ investigated in the present work, has been shown to form an antiparallel basket-type intramolecular quadruplex consisting of three stacked tetrads connected by two lateral loops and a central diagonal loop in Na${}^{+}$ solution [24]. The K${}^{+}$ structure, which is more relevant at physiological conditions, has also been intensively investigated. In particular, Tel22 crystal structure in the presence of K${}^{+}$ has been found to be parallel [25], while the conformation in K${}^{+}$ solution is a hybrid quadruplex with mixed antiparallel/parallel strands [26] (see Figure 1).

A considerable amount of work has been carried out to elucidate the structural properties of G4s and the effects upon ligand interaction [1,27,28,29,30,31,32,33,34]. However, much less is known about their dynamical properties, especially in the sub-nanosecond timescale. In proteins and nucleic acids, this dynamics has been shown to originate from jumps among conformational substates, i.e., nearly isoenergetic wells of their potential energy hypersurface [35]. We expect that the same concept can be applied to the case of G4s due to their intrinsic structural complexity. As it happens in the case of proteins [36,37,38], motions on the sub-nanosecond timescale can be linked to structural and functional aspects, as well as being precursor to slower conformational changes (polymorphism) and taking part to the early stages of ligand recognition [37,38].

In the present work, we investigated the picosecond timescale dynamics of Tel22 by measuring the atomic Mean Square Fluctuations (MSF) corresponding to internal biomolecular motions. The samples were studied in a hydrated powder state that, together with the use of deuterated solvent, allowed to obtain high signal-to-noise ratio for Elastic Incoherent Neutron Scattering (EINS) experiments. Moreover, the present water-poor conditions correspond to a highly crowded system that can possibly approximate molecular crowding in chromosomes. Two different Tel22 conformations were analyzed, obtained by stabilizing the G4 structure with either K${}^{+}$ or Na${}^{+}$ ions. Their topology was determined by Infrared (IR) spectroscopy measurements. The fast dynamics of Tel22 in the presence of these two ions was also studied upon complexation with BRACO19, which is a potential anticancer drug [39].

## 2. Results

### 2.1. Infrared Measurements

By Diffuse Reflectance Infrared Fourier Transform (DRIFT) experiments performed on all the dried samples, we first confirmed the formation of G4 structures by identifying a peak at 1485 cm${}^{-1}$, which is due to the guanine N7C8H bending vibration when the N7 atom is involved in Hoogsteen hydrogen bonds [40] (see Appendix A). As reported in Figure 2a, these measurements also indicate that the topology of the considered G4s tends towards a parallel conformation, as the guanine C6=O6 carbonyl stretching vibration peak is much closer to the value of 1692 cm${}^{-1}$ ascribed to the parallel topology than to the position associated to the antiparallel fold (1682 cm${}^{-1}$) [40].

The presence of the peak at 1484 cm${}^{-1}$ in the Attenuated Total Reflectance-Fourier Transform Infrared (ATR-FTIR) spectra shows that the G4 structures are preserved also at the investigated hydration level, as evidenced in Figure 2b. On the other hand, important conformational changes in the G4 topology occur in the hydrated samples. In particular, in Tel22 (K${}^{+}$) and Tel22-BRACO19 (K${}^{+}$) spectra, the guanine C6=O6 carbonyl stretching vibrational peak is located at 1691 cm${}^{-1}$, showing that the predominant conformation of G4s stabilized by K${}^{+}$ is the parallel one [40]. The same peak, however, is shifted towards lower wavenumbers in the presence of Na${}^{+}$, nominally at 1686 cm${}^{-1}$ in the case of Tel22 (Na${}^{+}$) and at 1688 cm${}^{-1}$ for Tel22-BRACO19 (Na${}^{+}$), suggesting that the Na${}^{+}$ stabilized G4s tend towards a mixture of antiparallel/parallel conformations [40]. As it turns out, complexation with BRACO19 does not seem to induce any major conformational variations (at least at the selected hydration level).

### 2.2. EINS Measurements

The MSF calculated through a Gaussian fit of the elastic intensity of Tel22 (K${}^{+}$), Tel22 (Na${}^{+}$), Tel22-BRACO19 (K${}^{+}$), and Tel22-BRACO19 (Na${}^{+}$) in the [1.0–4.6] Å${}^{-1}$
*Q* range are reported in Figure 3 as a function of temperature along with the best fits using Equation (Equation 4) to obtain the characteristic force constants (see Table 1). It is worth of note that, since the incoherent intensity in our samples is dominated by Tel22, with BRACO19 only contributing for 20% of the signal, the estimated MSF mainly represent the biomolecular internal dynamics. A marked change in the slope of the MSF was observed in the [225–235] K temperature range, likely related to a dynamical transition. This phenomenon, which has already been observed in other biological systems like proteins, is associated to the onset of anharmonic motions in the biomolecule, such confined reorientations of molecular groups [41,42]. These fluctuations are known to be coupled to the motions of the hydration shell water molecules [36]. As for the uncomplexed samples, at low temperatures the MSF are almost the same, while at higher temperatures Tel22 (Na${}^{+}$) displays smaller MSF than those obtained in the presence of K${}^{+}$, thus indicating that Na${}^{+}$ promotes a more stable quadruplex structure.

Quite interestingly, upon complexation with BRACO19, a similar difference is observed between the MSF of the K${}^{+}$ and Na${}^{+}$ stabilized G4s, as shown in Figure 4. A comparison of the effective force constants reported in Table 1 also shows that Tel22 (Na${}^{+}$) is characterized by a lower thermal flexibility than Tel22 (K${}^{+}$). Complexation with BRACO19 also leads to an increase of the MSF in both investigated environments, as shown in Figure 3, the increase in mobility being independent of the kind of ion (see Figure 4). The complexes display higher thermal flexibility than the samples in the uncomplexed state, as reported in Table 1. We remark that the effect of complexation on the magnitude of the MSF is larger than that of the different ions.

## 3. Discussion

Determining whether the G4 structures fold in a hydrated powder state and what topology they assume is not a trivial task. This is why here we employed DRIFT and ATR-FTIR techniques to monitor the formation of G4 structures and discriminate among their possible different topologies. We revealed that mostly parallel Tel22 G4s form in the presence of K${}^{+}$ ions, in close analogy with the case of Tel22 (K${}^{+}$) crystals already studied in the past by X-ray diffraction measurements [25]. This is also consistent with the finding that, in crowded K${}^{+}$ solutions, the parallel structure is stabilized [43]. On the other hand, Tel22 adopted a mixed antiparallel/parallel conformation in the Na${}^{+}$ powder state, which is consistent with the one adopted in Na${}^{+}$ crowded solutions [44]. The Tel22 parallel and mixed conformations, in the presence of K${}^{+}$ and Na${}^{+}$ respectively, were retained after complexation with BRACO19.

As for the binding mode of BRACO19 with Tel22, no high-resolution structures of BRACO19 are reported in the literature. MD simulations carried out on human telomeric G4 sequences have shown that the most stable and probable interaction occurs as an end, bottom, and top stacking for parallel, antiparallel, and hybrid G4 topology, respectively [45,46]. As these simulations were performed in the water-poor regime of concentrations comparable to the present hydrated powder conditions, it is reasonable that the results from the aforementioned studies can be applied to our systems.

Contrary to the case of hydrated powders, diluted solutions of Tel22 show the presence of a prevalently antiparallel topology after complexation with BRACO19, in either K${}^{+}$ or Na${}^{+}$ ionic environment (see Appendix A), as observed for other classic G4 stackers [47]. These results indicate that Tel22 self-crowding deeply alters the G4 conformation in both uncomplexed and complexed states, an effect that is probably related to the formation of higher-order multimeric G4 structures [44].

Based on the information derived from IR experiments, we can argue that the Tel22 mixed antiparallel/parallel structure is dynamically more stable than the parallel one in the sub-nanosecond timescale, in both the free (Figure 3c) and the complexed state (Figure 3d). Structural fluctuations in this timescale are coupled to solvent fluctuations [36,48]; therefore, it is reasonable to suppose that hydration water plays a role in the observed dynamical behavior of Tel22. It has been recently shown that extensive and stable spines of primary-sphere water molecules are formed in antiparallel and hybrid quadruplexes, but cannot be accommodated as easily in the less extended grooves of parallel G4 structures [49]. We can then hypothesize that these stable networks of ordered hydration water placed in the grooves of the antiparallel structural component—which is favored by the Na${}^{+}$ cations—are able to effectively hinder the G4 dynamics. In contrast, in the case of parallel G4 promoted by K${}^{+}$ ions, the presence of less ordered hydration networks reflects into an enhanced Tel22 fast dynamics [49]. The different behavior of Tel22 hydration water in the two different parallel and mixed antiparallel/parallel conformations is supported by the trend of the OH-stretching band (see Appendix A). Indeed, the intensity of the low wavenumber component associated to the so-called connective tetrahedral water (see Appendix A) increases for Tel22 powders with Na${}^{+}$ cations, thus suggesting that the amount of ordered water surrounding the antiparallel conformation is higher than that of the parallel conformation [23,50,51].

Regarding the effect of the BRACO19 ligand on Tel22 fast dynamics, we demonstrated that both mobility (MSF) and thermal flexibility ($k^{\prime }$) increase upon complexation. As shown in Figure 4, the resulting difference in MSF amplitude is independent of that induced by the ionic environment, thus indicating that these dynamic changes are uncorrelated with the aforementioned ones, which were already explained in terms of conformational variations. This behavior is consistent with the results obtained by FTIR analysis that, upon complexation, no or only minor conformational changes occur in the present crowded conditions.

On these grounds, it is reasonable to hypothesize that the increase of Tel22 MSF upon complexation with BRACO19 can be explained by a rearrangement of the hydration water network around the biomolecule rather than by a change of G4 topology [52]. As the magnitude of a biomolecule MSF above the dynamical transition temperature significantly increases with the hydration level [53,54,55], we can argue that larger MSF correspond to a higher number of water molecules interacting with Tel22. All the samples were prepared at the same hydration level *h* = 0.5 g D${}_{2}$O/g dry sample (Tel22 or Tel22-BRACO19 complex); however, we did not know in advance how hydration water would be distributed or how it would interact with Tel22 and BRACO19 components in the complexed state. We only knew that, if water molecules interacted in a similar way with both components, then the hydration level of Tel22 in the complex would be the same as that of unbound Tel22, i.e., *h* = 0.5 g D${}_{2}$O/g dry Tel22. Conversely, in the case of preferential hydration of either Tel22 or BRACO19, the hydration level of the biomolecule would be respectively larger or smaller than that of unbound Tel22. For instance, if we assume that no water molecules bind to the ligand, we will have an “effective” Tel22 hydration level *h* = 0.64 g D${}_{2}$O/g dry Tel22 in the complexed state. Therefore, given the observed increase of Tel22 mobility upon complexation, we propose that a preferential binding of water molecules to Tel22 against BRACO19 takes place. In this context, the excess of water molecules interacting with Tel22 in the complexed state triggers additional biomolecular degrees of freedom with respect to the uncomplexed one. We suggest that this preferential hydration can play a role in ligand binding to Tel22 by enhancing the entropic contribution to the binding conformational free energy. Our findings imply that the subtle interplay between Tel22 and its hydration water mediates the effect of both polymorphism and complexation with ligands on Tel22 fast dynamics. This picture complements results obtained from recent studies [56] showing that waters serve to keep the drug molecules in their stable binding position. As we showed that the Tel22 dynamics is significantly affected by the binding with BRACO19, future studies should focus on the effect of different kinds of small molecules, such as TMPyP2/4 and pyridostatin. Another promising example is Phen-DC3, which has recently been shown to induce a conformational change in human telomeric G4 from a hybrid 1 structure to a chair-type antiparallel one by intercalating between a two-quartet unit and a pseudo-quartet [47].

## 4. Materials and Methods

### 4.1. Sample Preparation

Lyophilized DNA oligonucleotide Tel22 sequence was purchased from Eurogentec (Seraing, Belgium). To obtain samples in K${}^{+}$ environment, the initial powder was dissolved in a 50 mM K-phosphate buffer at pH 7 containing 38.5% of KH${}_{2}$PO${}_{4}$ (monobasic salt) and 61.5% of K${}_{2}$HPO${}_{4}$ (dibasic salt), 0.3 mM EDTA, and 150 mM KCl. For Na${}^{+}$ samples, the initial powder was dissolved in a 50 mM Na-phosphate buffer at pH 7, containing 38.5% of NaH${}_{2}$PO${}_{4}$ (monobasic salt) and 61.5% of Na${}_{2}$HPO${}_{4}$ (dibasic salt) 0.3 mM EDTA, and 120 mM NaCl. Each sample was lyophilized again, dissolved in D${}_{2}$O at a concentration of about 50 mg/mL, and left at room temperature for 1 day to allow D-H substitution. The solution was then freeze-dried and dehydrated under vacuum in the presence of P${}_{2}$O${}_{5}$. The dried powder was finally hydrated with D${}_{2}$O, until a hydration level *h* = 0.5 g D${}_{2}$O/g dry Tel22 was achieved. In duplex DNA, water molecules are strongly coordinated by phosphate groups up to at least *h* = 0.6 g D${}_{2}$O/g dry DNA. By analogy, we expect no free solvent in G4s at the investigated hydration level.

BRACO19 ligand was purchased from Merck KGaA (St. Louis, MO, USA) and directly dissolved in K/Na phosphate buffer. Its molecular structure is shown in Figure 1c. The oligonucleotide samples above described were complexed with the ligand in 1:2 [G4]:[ligand] stoichiometric molar ratio. Samples were left for two hours at room temperature in order to reach complexation. Hydrated powders of Tel22-BRACO19 complexes at a hydration level of *h* = 0.5 g D${}_{2}$O/g dry Tel22-BRACO19, both in K${}^{+}$ and Na${}^{+}$ environments, were obtained following the same procedure discussed above.

The samples were again dehydrated under vacuum for DRIFT measurements and subsequently hydrated with H${}_{2}$O to a hydration level *h* = 0.5 g H${}_{2}$O/g dry sample for ATR-FTIR measurements.

### 4.2. Neutron Scattering Experiments and Theoretical Model

Incoherent Neutron Scattering (INS) experiments are a key tool in the investigation of fast fluctuations in biomolecules. The scattering of thermal neutrons from hydrogen atoms, which is prevalently incoherent in nature, is by far larger than that from any other element and isotope in biological samples. As a consequence, measuring D${}_{2}$O hydrated G4 powders allows to neglect the signal arising from the solvent, so that only non-exchangeable hydrogen atoms of the biomolecule significantly contribute to the detected signal [57,58]. The measured intensity allows to directly calculate the incoherent dynamic structure factor S(Q,E), where *E* and ℏQ are the energy and the momentum transfer, respectively. Therefore, information on the self-particle dynamics of hydrogen atoms that are almost uniformly distributed within the G4 can be obtained, thus allowing to probe the fast motions of larger groups of atoms to which hydrogens are bonded.

In the present study, INS experiments were performed on the thermal neutron back-scattering spectrometer IN13 at the Institut Laue-Langevin (ILL), with an energy resolution of ≈8
μeV full width at half maximum in the large wavevector coverage [0.2–4.9] Å${}^{-1}$. CaF${}_{2}$(4,2,2) crystals were used as monochromator and analyzers in back-scattering geometry. The IN13 energy resolution corresponds to an accessible time window of up to 150 ps. Neutrons were counted with a cylindrical multidetector consisting of 35 ${}^{3}$He tubes and with a ${}^{3}$He Position Sensitive Detector (PSD), which was used to cover the [0.2–0.8] Å${}^{-1}$ Q-range [59]. Elastic scans from 20 K to 300 K were carried out for all the samples. Each of them, consisting in 100 mg of powder, was filled in sealed Aluminium flat cans and kept in a cryostat during the acquisition. We remark that, in the present case of powder samples, only the biomolecular internal dynamics is probed, as the translational and rotational degrees of freedom are suppressed in this timescale. In the case of complexed samples, we calculated the maximum contribution to the incoherent and total signal arising from BRACO19, which resulted to be 22% and 19% of the detected signal, respectively. The sample transmission of the measured samples was in the 0.92–0.95 range. The standard reduction of raw data (empty can subtraction, sample transmission correction, and normalization by the lowest temperature intensity) was carried out using the LAMP Software [60].

EINS experiments consist in measuring the elastic intensity, i.e., the number of neutrons that are scattered from the sample with an energy transfer lying within the energy resolution of the spectrometer [57]. The measured elastic intensity can be expressed as:(1)$$I\left(Q,\omega =0\right)=\sum_{\alpha =1}^{N_{\alpha }}x_{\alpha }\langle e^{i\vec{Q}\cdot \left[{\vec{r}}_{\alpha }\left(\infty \right)-{\vec{r}}_{\alpha }\left(0\right)\right]}\rangle =\sum_{\alpha =1}^{N_{\alpha }}x_{\alpha }\langle e^{i\vec{Q}\cdot {\vec{u}}_{\alpha }}\rangle$$
where $x_{\alpha }$ is the fraction of dynamically equivalent hydrogen atoms whose position vector at time *t* is denoted by ${\vec{r}}_{\alpha }\left(t\right)$. A Gaussian approximation to this quantity can be written as:(2)$$I\left(Q,\omega =0\right)≃I_{0}e^{-\frac{1}{6}Q^{2}\langle u^{2}\rangle }$$
where
(3)$$\langle u^{2}\rangle =\sum_{\alpha =1}^{N_{\alpha }}x_{\alpha }\langle u_{\alpha }^{2}\rangle$$
is the atomic Mean Squared Displacement (MSD) averaged over all hydrogen atoms and $I_{0}$ is the elastic intensity extrapolated at *Q* = 0. This approximation holds in the limit $Q^{2}\langle u^{2}\rangle <2$ [61,62,63].

Our EINS data were collected in the wide *Q*-range accessible to the IN13 spectrometer, which is usually too large for the Gaussian approximation to be valid [64,65,66]. However, Equation (Equation 2) provides an excellent fit to the elastic scattered intensity, as shown in Figure 5, thus suggesting that G4 hydrogen atoms undergo small amplitude fluctuations in the investigated timescale. MSF $\langle r^{2}\rangle$ from the equilibrium position, which are half of the MSD obtained from Equation (Equation 3), can be used to assess the thermal flexibility of the system [61]. As already discussed, the system is characterized by a complex multidimensional energy landscape. Therefore, its dynamics can be modeled by considering the scattering hydrogen atoms as random walkers moving among conformational substates in a confining potential. At low temperature, their thermal fluctuations are relatively small, so that only a limited part of the landscape is explored and their motions mainly consist of harmonic oscillations around local conformational minima. At higher temperature, however, a larger portion of the landscape can be explored with scatterers crossing energy barriers between local minima and hence activating anharmonic motions. Assuming that the confining potential is quadratic, we obtain the following temperature dependence for the MSF [36]:(4)$$\langle r^{2}\rangle =3\frac{k_{B}T}{k^{\prime }}$$
where $k_{B}$ is the Boltzmann constant and $k^{\prime }$ is an effective *conformational* force constant, which properly quantifies the resilience of the scatterers against the jump among conformational minima. $k^{\prime }$ provides a measure on the way fluctuations in the investigated timescale contribute to the system thermal flexibility.

### 4.3. Infrared Spectroscopy

ATR-FTIR measurements were performed in order to investigate the conformational structure of our samples. Indeed, peculiar vibrational peaks visible in IR spectra can be assumed to be clear markers of different G4 conformations [40,67]. ATR-FTIR spectra were acquired using a FTIR Bruker spectrometer (mod. Alpha-P), equipped with a Platinum ATR module employing a single reflection (45°) diamond crystal. Each spectrum was obtained at room temperature, averaging over 120 scans at a resolution of 2 cm${}^{-1}$, using the spectrum of the empty ATR plate as the reference background. Spectral profiles in the [380–5000] cm${}^{-1}$ frequency range were analyzed using the *Baseline Correction* routine implemented in the Opus 7.5 Bruker Optics software. Measurements were carried out on water hydrated powder samples.

DRIFT spectra were acquired with a Bruker IFS66V Michelson interferometer (Bruker Optik, Ettlingen, Germany), working under vacuum and with a spectral resolution of 2 cm${}^{-1}$. Each spectrum was the average of 256 single acquisitions: this resulted in a noise-to-signal ratio lower than 1%. The method adopted for the evaluation of the absorption coefficient was based on the measure of the diffuse infrared reflectivity. This procedure detected the diffuse reflectance R(ω) of a blend composed by sample powder dispersed in dried KBr (5% *w*/*w*). Afterwards, Kubelka-Munk transformation of the diffuse reflectance provided the absorption coefficient A(ω) of the sample in the mixture through the following relation:(5)$$A=\frac{{\left(1-R\right)}^{2}}{2R}$$
Further corrections for the residual atmospheric absorption and baseline mismatch were carried out with the OPUS code for data analysis. Measurements were carried out on dehydrated powder samples.

## Figures and Tables

**Figure 1 ijms-24-04280-f001:**
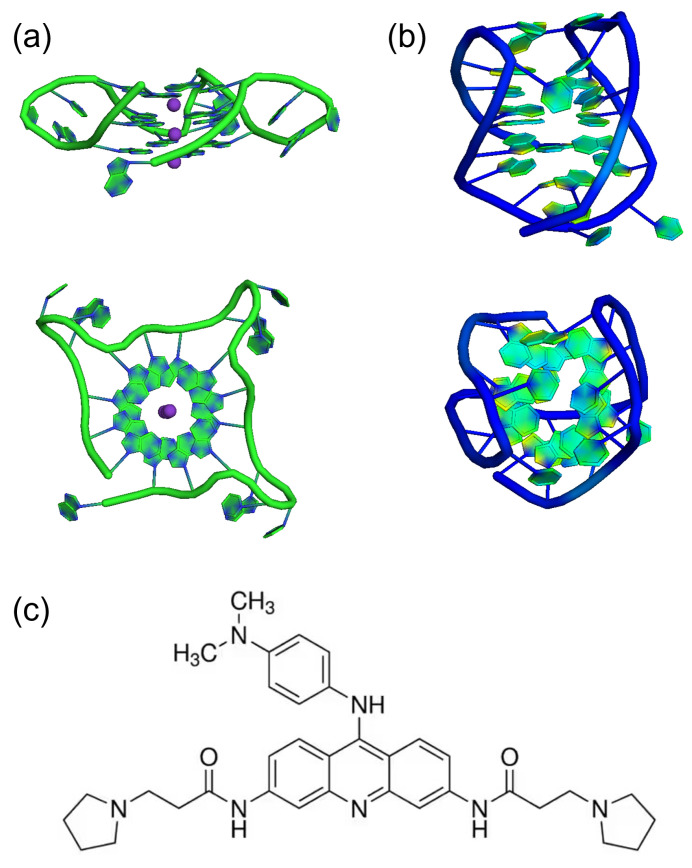
Cartoon representation of the PDB structures of AG${}_{3}$(TTAG${}_{3}$)${}_{3}$ (Tel22) with different cations: (**a**) Tel22 crystal structure in K${}^{+}$, PDB 1KF1 [25], (**b**) Tel22 structure in Na${}^{+}$, PDB 143D, as obtained from nuclear magnetic resonance experiments [24]. (**c**) Molecular structure of BRACO19.

**Figure 2 ijms-24-04280-f002:**
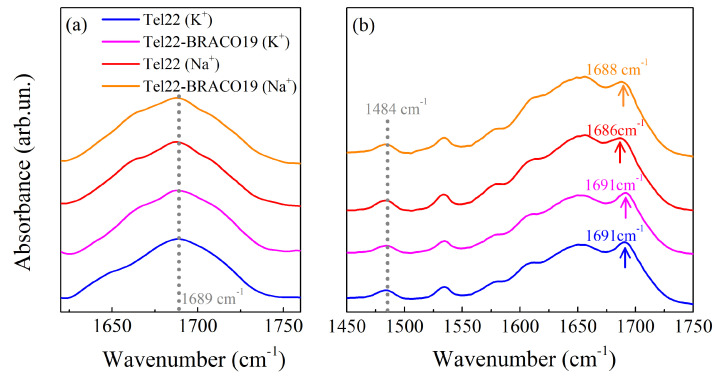
(**a**) Diffuse Reflectance Infrared Fourier Transform (DRIFT) spectra of of Tel22 (K${}^{+}$), Tel22 (Na${}^{+}$), Tel22-BRACO19 (K${}^{+}$), and Tel22-BRACO19 (Na${}^{+}$) in the C6O6 carbonyl stretching vibrational peak region, highlighted with dotted line. (**b**) Attenuated Total Reflectance-Fourier Transform Infrared (ATR-FTIR) spectra of the same four samples in the [1450–1750] cm${}^{-1}$ spectral region. The arrows indicate the peak center for each spectrum. The dotted line represents the G-quadruplex (G4) characteristic peak.

**Figure 3 ijms-24-04280-f003:**
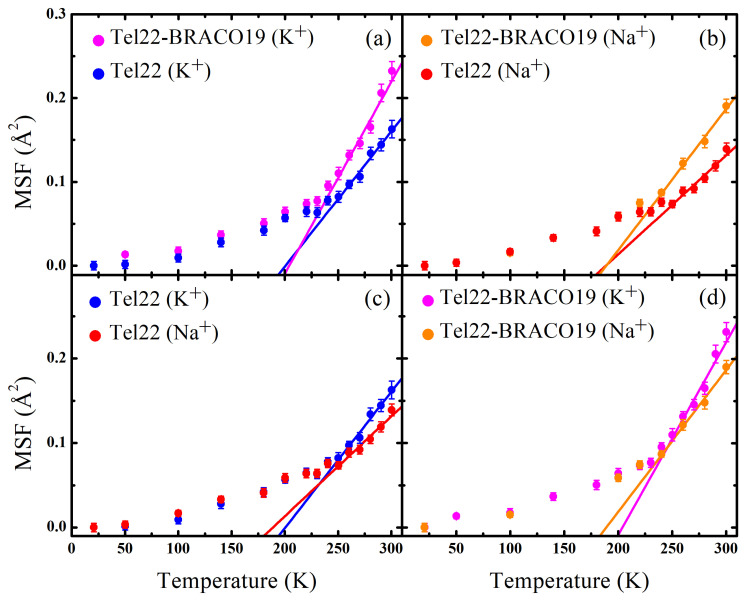
Mean Square Fluctuations (MSF) as a function of temperature for Tel22 (K${}^{+}$), Tel22 (Na${}^{+}$), Tel22-BRACO19 (K${}^{+}$), and Tel22-BRACO19 (Na${}^{+}$) along with the best fit to data (solid lines) using Equation (Equation 4).

**Figure 4 ijms-24-04280-f004:**
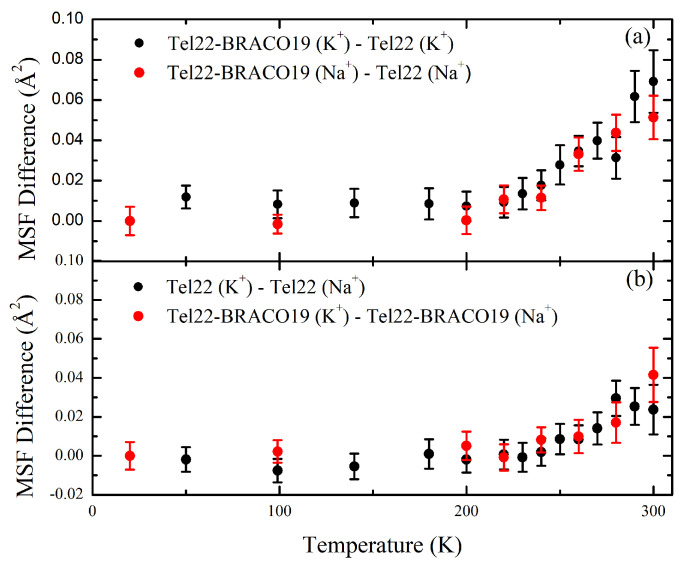
Comparison of MSF of different samples as a function of temperature, considering the effects induced by complexation (**a**) and by ionic environment (**b**).

**Figure 5 ijms-24-04280-f005:**
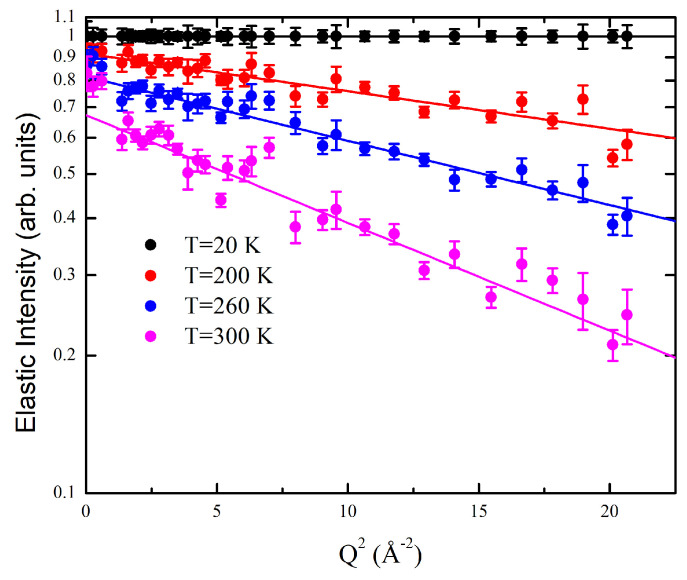
Normalized elastic scattered intensity (logarithmic scale) as a function of $Q^{2}$ for Tel22 in the presence of K${}^{+}$ ions at four different temperatures. In this scale the Gaussian fit using Equation (Equation 2) over the [1.0–4.6] Å${}^{-1}$
*Q* range appears as a linear plot.

**Table 1 ijms-24-04280-t001:** Force constants as extracted from the best fit to the data using Equation (Equation 4).

	Tel22 (K${}^{+}$)	Tel22 (Na${}^{+}$)	Tel22 (K${}^{+}$)	Tel22 (Na${}^{+}$)
			-BRACO19	-BRACO19
$k^{\prime }$ (N/m)	2.9 ± 0.2	3.6 ± 0.3	2.1 ± 0.3	2.7 ± 0.1

## Data Availability

Not applicable.

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
