# Peer review of "Polymorphism and Ligand Binding Modulate Fast Dynamics of Human Telomeric G-Quadruplexes"

_ijms, 2023, doi:10.3390/ijms24054280_

Round 1

Reviewer 1 Report

The study is interesting and meaningful. Authors’ experimental results showed that the presence of ordered hydration water networks could influence G4 conformation and stability. However, there are still issues to be addressed, for example, is there a need for in-depth understanding the conformation changes of G4 by the co-crystal technique? Therefore, present manuscript is not suitable for publication in Int. J. Mol. Sci at this stage. In addition, there are following questions to be answered:

1.     In order to explain conformation changes in powder/diluted samples, CD spectra of Tel22 G4 in K+/Na+-containing diluted solutions should also be provided.

2.     Authors should sufficiently explain that there are almost no or only minor conformational changes of G4 upon complexation of G4s with BR19. Also, what is the interaction mode of BR19 with G4s under powder conditions?

3.     P2, line 72-74: “--- The powder was dissolved in a 50 mM K-phosphate buffer at pH 7, 0.3 mM EDTA, 150 mM KCl for the K+ samples and in a 50 mM Na-phosphate buffer at pH 7, 0.3 mM EDTA, 120 mM NaCl for the Na+ ones.” Author should more clearly express the composition of buffer.

4.     It is suggested that authors show the structures of Tel22 G4 in the presence of K+/Na+ or BR19 and that of BRACO19 (BR19).

Reviewer 2 Report

Overall, this paper studies the polymorphism and the ligand-induced modulation of dynamics of human telomeric G-quadruplexes (G4s). The research reveals the influence of ions on G4 topology by different biophysical assays, also studies the complexation with BRACO19 ligand. This work is worthy of publication in IJMS. However, the following concerns should be addressed before publication. In addition, the editing of the manuscript should be improved.

Major concerns:

1. There is always a concern that the samples were tested in powder/dry state which might give different conclusions from the in-solution results. Please explain or highlight the significance of the study of hydrated samples.

2.  Aside from BRACO19, other classic G4 stackers have also been developed, such as TMPyP2/4, Phen-DC3, pyridostatin (PDS), etc. Among them, Phen-DC3 has recently been shown to induce the refolding of human telomeric DNA into a chair-type antiparallel G4 structure (Angewandte Chemie, https://doi.org/10.1002/ange.202207384). It clearly reveals that different ligands can differently affect the topology of G4, which should be discussed in your manuscript.

3. About the BRACO19 tested in this study, normally the small molecule is dissolved in DMSO for most biophysical/biological assays. How did you remove the solvents? Or did the authors use the salt version and dissolved in water? Please clarify in the method part.

Minor concerns:

1.     Please remove the abbreviation section in the manuscript (page 9). Instead, please make sure to put the full name plus corresponding abbreviation in the main text when it is introduced for the first time. Besides, there is no need to add an abbreviation (BR19) for BRACO19. Most literature related to G4 study directly use BRACO19 (or BRACO-19).

2.     In abstract, line 4, it should be Fourier transform infrared spectroscopy (FTIR).

3.     The symbol of “~” in the description of ranges are wrong, which might be due to the document display (see page 3, line 105~108; page 4, figure 1 legend). Please check the format.

Round 2

Reviewer 1 Report

Authors have answered all questions.

Reviewer 2 Report

The revised the manuscript has already addressed the concerns in the previous review comments. The description on the data matches the experimental design. In addition, more details have been added in the result section, which provides more comprehensive discussion. Overall, the revised manuscript is suitable for publication in IJMS.